# Minimally Invasive Retrosigmoidal Parasterional Burr-Hole Approach: Technique and Neuropathic Pain Amelioration after Microvascular Decompression of the Trigeminal Nerve

**DOI:** 10.3390/biomedicines11102707

**Published:** 2023-10-05

**Authors:** José Damián Carrillo-Ruiz, Juan Camilo Covaleda-Rodríguez, José Armando Díaz-Martínez, Antonio Vallejo-Estrella, José Luis Navarro-Olvera, Francisco Velasco-Campos, Armando Armas-Salazar, Fátima Ximena Cid-Rodríguez

**Affiliations:** 1Unit of Functional Neurosurgery, Stereotactic and Radiosurgery, General Hospital of Mexico, Mexico City 06720, Mexico; camilocovaleda@hotmail.com (J.C.C.-R.); dr.armandodiaz@gmail.com (J.A.D.-M.); ave0005@gmail.com (A.V.-E.); luiginavarro97@hotmail.com (J.L.N.-O.); slanfe39@gmail.com (F.V.-C.); armandoarmassalazar@yahoo.com.mx (A.A.-S.); fatimaximena.cid@upaep.edu.mx (F.X.C.-R.); 2Coordination of Neuroscience, Faculty of Psychology, Mexico Anahuac University, Mexico City 52786, Mexico; 3Research Direction, General Hospital of Mexico, Mexico City 06720, Mexico; 4Unit of Functional Neurosurgery and Stereotactic, Olaya Polyclinic Center, Bogota 111411, Colombia; 5Department of Neurosurgery, Hospital Universitario “Dr. José Eleuterio González”, Monterrey 64460, Mexico; 6Postgraduate Department, School of Higher Education in Medicine, National Polytechnic Institute, Mexico City 07360, Mexico

**Keywords:** retrosigmoid approach, parasterional, burr-hole, trigeminal neuralgia, surgery, minimally invasive

## Abstract

Background: Trigeminal neuralgia, a common condition in clinical practice, often occurs due to vascular compression caused by aberrant or ectopic arterial or venous vessels. Microvascular decompression through a minimally invasive retrosigmoidal approach has shown high rates of pain control, low complication rates, and excellent therapeutic results. Objective: To describe the surgical technique and clinical outcomes in terms of pain relief after microvascular decompression of the trigeminal nerve through a minimally invasive retrosigmoidal parasterional burr-hole technique. Methods: A group of patients with trigeminal neuralgia refractory to medical management who underwent microvascular decompression were examined. The records of the patients were considered retrospectively (2016–2018), and the outcomes were considered based on the Visual Analogue Scale (VAS) and the Barrow Neurological Institute Pain Scale (BNIPS) added to a technical note of the surgical technique for a minimally invasive retrosigmoidal parasterional burr-hole. Results: Twenty-two patients were evaluated, and clinical assessment after surgical intervention showed a decrease in pain according to the VAS, resulting from an average preoperative state of 9.5 ± 0.37 to a postoperative condition of 1.32 ± 1.28, exhibiting statistically significant changes (*p* < 0.0001, d = 9.356). On the other hand, in relation to the BNIPS scale, a decrease from an average preoperative status of 4.55 ± 0.25 to a postoperative status at 12 months of 1.73 ± 0.54 was also demonstrated, showing significant changes (*p* < 0.0001, d = 3.960). Conclusion: Microvascular decompression of the trigeminal nerve through a minimally invasive retrosigmoidal parasterional burr-hole is feasible and can be a safe and effective technique for the management of pain. However, further research employing larger sample sizes and longer follow-up periods is necessary.

## 1. Introduction

Trigeminal neuralgia (TN) is defined as a neurological disorder characterized by paroxysmal pain, which is generally associated with trigger points that correspond to areas innervated by a branch of the trigeminal nerve; this pain can occasionally be manifested continuously, and it is a relevant factor for disability during the disease and the functional limitation associated with it [1]. TN can be associated with ipsilateral facial spasms (tic douloureux) and mild autonomic symptoms such as epiphora and/or ipsilateral conjunctival injection. After a painful paroxysm, which is generally short, there is a refractory period, during which pain is not triggered [2]. TN is, in most cases, a neurovascular compression in the trigeminal root entry zone. This compression can lead to the demyelination and dysregulation of voltage-gated sodium channel expression in the membrane (alterations responsible for pain) [3]. The proposed chain of events leading to neurovascular compression-related symptoms is as follows: vascular compression in the transition zone → demyelination → nucleus hyperexcitability → symptoms [4]. According to the beta version of the international classification for headache disorders (ICHD-3 Beta 2018), for its diagnosis, it should be defined with the presence of unilateral paroxysmal pain, clearly delimited to a division of a trigeminal nerve branch, associated with allodynia, and not explained by an alternate diagnosis [2]. 

Once the clinical diagnosis of TN is established, it should be complemented with an MRI assessment in a multi-planar projection with thin slices and an SSFP sequence with great T2 weighting, which provides excellent contrast resolution. Neuroimaging plays a pivotal role in differential diagnosis in patients with non-vascular causes (tumors, aneurysms, arteriovenous malformations, among others). It is important to note that while many neurovascular causes can be observed intraoperatively, they may not be visualized using neuroimaging methods [3]. Therefore, in some cases, intervention is justified based primarily on the symptoms in the absence of an imaging finding. SSFP sequences are often mentioned using their specific abbreviation given by the provider [5], either FIESTA (fast imaging employing steady-state acquisition) or 3D CISS (three-dimensional constructive interference in steady state), to investigate possible vascular etiologies, mainly neurovascular compression of a nerve, such as the superior cerebellar artery, anteroinferior cerebellar artery, basilar artery, and venous complex, as well as other causes such as tumoral lesions [6]. 

The annual incidence of TN has been reported at a rate from 4.7 to 12.6 per 100,000 people [7,8], with a predominance in females, where 51 years is the average age of presentation [8]. Initial treatment for TN is pharmacological, and carbamazepine is the drug of choice, but oxcarbazepine, pregabalin, lamotrigine, phenytoin, topiramate, and amitriptyline are also useful, and in some cases combined schemes are required to obtain therapeutic results [9]. When pharmacological treatment fails, there are a variety of procedures that can be applied, which include microvascular decompression (MVD), percutaneous radiofrequency rhizotomy, percutaneous rhizotomy with glycerol, balloon percutaneous compression, and stereotactic radiosurgery [10].

Traditionally, MVD of the trigeminal nerve has shown a rate of success in the management of pain between 80.3 and 96%, with the immediate control of pain and variable rates of complications that oscillate between 4 and 6% [11,12,13], and this process involves performing a conventional approach through a retrosigmoid craniotomy, making a keyhole of approximately 40 mm with a sufficient and wide exposition. 

We present the technique of a retrosigmoid approach through a miniasterional craniotomy as an effective and safe technique that allows a proper surgical corridor for the upper, middle, and lower cerebellar complexes in case this is required for associated pathologies [14]. 

## 2. Methods and Materials

A total of 22 patients were treated at the neurosurgery Department of the General Hospital of Mexico (Mexico City, Mexico) for the clinical management of TN. The records of patients managed in our clinic were considered (2016–2018). According to the eligibility criteria, all adult patients (18–65 years) of both genders, with TN secondary refractory to medical management, anatomic impairment of the fifth nerve (determined through magnetic resonance imaging) were included. Those patients with previous surgical intervention for TN, evidence of multiple sclerosis or other neurological disorders that may mimic TN, and contraindications to surgery, pregnancy, or lactation were excluded. The data extraction was focused on collecting information on the demographic aspects (age, gender), etiology, anatomical location/distribution of the nervous assault, and the affected side. The clinical evaluation of the patients was focused on collecting data corresponding to the pre- and postoperative state of the pain status according to the Visual Analogue Scale (VAS), and the Barrow Neurological Institute Pain Scale (BNIPS). Statistically significant differences between pre-operative and post-operative pain scales were calculated using an ANOVA test and post hoc analysis with DMS or C-Dunnett tests, depending on the homogeneity of the variances defined through Levene’s test. The effect size was calculated using Cohen’s d and recalculated considering the correction coefficient for small sample sizes. Data measures were performed using SPSS 25.0 for Windows software (SPSS, Inc., Chicago, IL, USA), where a *p*-value < 0.05 was considered significant.

### Technical Note

Proper positioning is thought to be the cornerstone for performing a minimally invasive retrosigmoidal parasterional burr-hole approach, since it avoids the retraction of the cerebellum and prevents contralateral venous return from alteration, which is of great help and utility at the moment of the procedure (Figure 1A–C). A 3-pin cephalic support is placed (Doro-Integra LifeSciences Corporation, Plainsboro, NJ, USA). The patient is positioned in lateral decubitus, contralateral to the area being approached, with the inferior arm outside the surgical table, held over a sling. The chest is elevated approximately 15°, and the head is placed with a minimum flexion of 5° and a 10° inferior rotation, so that the retromastoid plane is parallel to the floor. The park bench position is preferred over the supine decubitus approach with cephalic rotation, for providing a better venous contralateral return; control of the upper, middle, and lower cerebellar complexes; and being more comfortable for the neurosurgeon [15].

Relative to the incision, a retro-auricular curved incision is performed, 50 mm behind the tragus, starting from the upper portion of the mastoid, 5 mm in front of the digastric cleft, and is extended 10 mm above where the asterion has been referred. To locate the position of the asterion, Reid’s orbitomeatal line is projected towards the posterior, which corresponds in all cases to the base of the zygoma and the union of the upper third with the medial third of the auricular pavilion. The asterion is located where Reid’s line crosses with a line drawn up to the digastric cleft (Figure 2A,B). An incision is made, approximately 50 mm long, and subperiostic dissection is made with monopolar, revealing the digastric cleft, the mastoid base towards inferior and the asterion towards superior [16]. Retro mastoid emissary vein is controlled with bone wax and is not considered an anatomic reference for the craniotomy, given the great variability of its position [17].

Regarding craniotomy, a single keyhole is made with a self-blocking drill bit, 14 mm below the union of the sigmoid and transverse sinuses, medial and inferior from the asterion (Figure 2C,D); identification of the transverse and sigmoid sinuses is necessary for a safe dural opening [18]. Different from the conventional retrosigmoid approach, we consider a parasterional craniotomy (<20 mm) as sufficient for the cerebellopontine angle exploration to handle vascular compressions associated with TN. This approach not only allows manipulation of the upper cerebellar complex, but also provides an excellent surgical corridor for the middle and lower ones [14,19].

In relation to dural opening, it was performed in a triangular shape, with a base of approximately 7 mm, in direction towards the sigmoid sinus, in order to have only one reference to avoid dural retraction (Figure 2E,F). This kind of dural opening is more comfortable for closure, minimizes the requirement for an autologous graft, and reduces the risk of unnoticed injury of the transverse and sigmoid sinuses at the time of closure. 

The approach to the cerebellopontine cistern was performed through passive drainage of cerebrospinal fluid, which can be maximized via contralateral rotation. This maneuver also allows visualization of the posterior petrous surface and the pathway to access the cistern of the cerebellopontine angle. Under magnified vision, the petrous surface is followed with the centered microscope and an angle of approximately 110–120° for adequate advancement in the angle’s cistern (Figure 3A,B); liberation of arachnoid adhesions is needed for the cerebellar hemisphere to drop without retraction. The first step consists of identifying the Tübingen line, which indicates the suprameatal tubercle projection [20], demonstrating the subarcuate artery and an arachnoid layer that covers the middle cerebellar complex [21]. To approach the trigeminal nerve, an arachnoid opening is made above the middle cerebellar complex. The microscope is angled 20° laterally to visualize the upper portion. Between the superior petrous venous complex (Dandy’s vein) and the middle cerebellar complex, the trigeminal nerve is identified in its cisternal portion; in our experience, we consider it not necessary to handle or coagulate Dandy’s vein in this work angle. Although the incidence of complications due to Dandy’s vein obliteration is low, it can occur, and sequels could be worse for the evolution of the existing pathology [22,23]. The trigeminal nerve is identified by its larger size compared to the middle cerebellar complex; a motor branch is identified at the nerve’s upper aspect. Release of arachnoid adhesions should be performed in an organized manner along the nerve, with special caution on the medial surface. We recommend always exploring from the inferior edge of the nerve to avoid handling the motor branch and to understand the trigeminal nerve with a somatotopic organization, lowering the injury possibility of branches in V1. In cases where a vascular association is not observed as the cause of compression, strong arachnoid reactions can be identified over the nerve, with good results after its complete liberation from the nerve. The superior cerebellar artery has been identified as the most frequent cause of compression, followed by venous compression and the anterior–inferior cerebellar artery (Figure 4A–D).

The release of the cisternal portion of the trigeminal nerve should be systematic, checking all its aspects according to the detailed planning with the MRI in high resolution sequences. Once we have achieved the vascular liberation of the trigeminal nerve, we proceed to implant a polytetrafluorethylene interface (PTFE) which can be found in diverse presentations (Figure 4E).

Dural closure was carried out with the dura mater opening in a triangular shape. The closure becomes easier, and the risk of fistulae is minimized, since it does not retract. With the point of reference located in the flap apex, closure is commenced with a simple continuous suture. Valsalva mechanisms are performed to document a hermetic closure and the absence of cerebrospinal fluid fistulae.

## 3. Results

Twenty-two patients were considered in the study. The affected population had a mean age of 54.9 ± 12.73, with women comprising 81% of the cases. The right side was predominantly affected (86%). The most common cause was vascular contact of the superior cerebellar artery, accounting for 59% of cases; branches V2 and V3 of the trigeminal nerve are predominantly affected (58%) (Table 1). Regarding clinical outcomes, we observed that, according to the VAS, there was a decrease in pain intensity from an average preoperative value of 9.5 ± 0.37 at 12 months to a postoperative value of 1.32 ± 1.28, demonstrating significant changes between the 12-month postoperative evaluation (*p* < 0.0001) with a considerable effect size (d = 9.356). On the other hand, in relation to the BNIPS scale, a decrease from an average preoperative status of 4.55 ± 0.25 to a postoperative status at 12 months of 1.73 ± 0.54 was also demonstrated, showing significant changes (*p* < 0.0001, d = 3.960) (Figure 5). Relative to the distribution of patient’s pain status, 63% of the population had a VAS score of 10 in the preoperative period, and after surgery (12 months after surgery), 68% of the sample presented a VAS equal to 0 (Table 2). Regarding the BNIPS, 55% of the sample presented a score of V before surgery, which after the intervention and the evaluation at 12 months, showed that most of the patients were found to have a score of I (63%) (Table 3).

The post-surgical complications observed in the patients were as follows: Four individuals experienced paresthesia in the territory of the decompressed trigeminal nerve, all of whom were classified as having mild symptoms that did not affect their daily activities. There were two cases of cerebrospinal fluid fistulas, and one patient developed a post-surgical hematoma that required further intervention. During the follow-up, no infections were found. Among the remaining 15 patients, there were no post-surgical complications.

## 4. Discussion

Neuropathic pain is the main indication for MVD in TN. According to a recent systematic review published by Holste et al. (2019), 46 articles were included from 1988 to 2018 in which only 28% (13 studies) reported pain using a standardized clinical scale [24]. The most frequent scales applied were BNIPS and VAS. The rest of the articles used a non-standardized tool to measure the outcomes of their patients, and some omitted them [24].

In the beginning of the TN surgery, Dandy used to make big craniectomies to approach the cerebellopontine angle (≈7 × 5 cm) [25]. Nowadays, the use of smaller incisions is becoming more popular in the decompression of TN [26]. Nevertheless, there is confusion in the terminology used for the denomination of the bone removal; for example, the terms “craniotomy”, “craniectomy”, “burr hole”, and “key hole” are commonly applied to refer to the approach without defining the size of this aperture. The current trend in surgical management for TN is to make smaller holes, including using endoscopic techniques. We believe that the name of the surgery should be standardized to avoid confusion; we propose the term retrosigmoidal parasterional burr-hole approach after considering the previously mentioned aspects. The use of reduced approaches has the purpose of reducing the damage on the scalp of the patient while still having an appropriate amount of space for the neurosurgeon to perform the surgery without inconvenience. 

According to a systematic review focused on hearing loss after MVD published by Bartindale in 2017, 35 studies reported hearing loss (in minimally invasive procedures) [27]. Between the reports of the two systematic reviews, one focused on pain (Holste, 2019) and the other on hearing loss (Bartindale, 2017) [27]. We found the following results: Only 28% (23 studies of 81) of the studies reported the size of the bone removal; the term most commonly used to report the surgical approach was “Craniectomy” in 30% of the articles (25 studies of 81), and the average size of the bone removal was not always the same in the studies. The range size reported for the term “craniectomy” according to this study was from 15 to 30 mm; for this reason, the term craniectomy is misused because in the rest of the literature it refers to approaches bigger than 40 mm [24,26].

The use of the terminology to describe the surgical approach previously mentioned has the purpose of describing a small hole on the scalp of the patient, but the area applied should be standardized with the aim of establishing relations between the space of the surgery and the clinical improvement. However, a craniectomy is not an adequate term for the procedure, because an article that evaluates the trephinations, trephines, and craniectomies found that “craniectomy” is a term used to refer to a huge cranial opening or window of variable shape that may include a size bigger than 4 cm that corresponds to the size of a trepan [27]. For this reason, it was decided to refer to this technique as a retrosigmoid parasterional burr-hole, because the description of a burr-hole has an average size of a dime (approximately 17.9 mm), the same dimensions used in the patients included in the present study (18 mm) [28]. 

TN is a high-incidence pathology with high health costs. MVD of the trigeminal nerve is established as a therapeutic strategy, with complete resolution of symptoms and low remission in 10 years of follow up [13]. Historically, the first case of MVD was performed by Gardner and Miklos, using a transtentorial subtemporal approach [29]. Later in 1967, Janneta [30], from microsurgical observations, developed the pathophysiological bases that support the arterial compression of the trigeminal nerve, establishing MVD through a retrosigmoid approach as the standard treatment [17]. Minimally invasive modifications of the technique have been previously described and popularized by several authors in recent years [17,31,32]. More recently, pure or combined endoscopic approaches have been described for treating TN, showing superiority in the microsurgical technique with visual improvement as well as better illumination of nervous and vascular structures, also being minimally invasive [33]. Nevertheless, the size of craniotomy proposed by these groups is using a 2.7 mm endoscope, so, with the use of a 4 mm conventional endoscope, the cranial opening is like the one we propose [34]. According to our knowledge, there is no description of a technique for a minimally invasive approach for MVD of the trigeminal nerve. The retrosigmoid parasterional burr-hole approach is proposed as an effective surgical option for the management of this pathology, minimizing cerebellar manipulation, and with low complication rates, microsurgical training is recommended for avoiding unnecessary injuries to vital structures, having the expertise of the cerebellopontine angle surgical anatomy. MVD must correspond to a safe surgery, with results dependent on a proper trigeminal nerve release, without forgetting any step that guarantees good surgical outcomes.

We consider the parasterional burr-hole as an adequate pathway for the management of the cerebellopontine angle in a posterior normal-tension fossa, at the expense that successful MVD is based on the release of the cisternal portion of the trigeminal nerve from the arterial or venous aberrant or ectopic vessel causing the pathology. Thus, a clear anatomy of the area allows for rapid identification of the work pathway, avoiding conflictive areas such as the superior petrous venous plexus and the facial nerve, shortening surgical time. It must be emphasized that for a successful MVD with this approach, adequate positioning of the head is needed for attaining the vision angles required for not retracting the cerebellum, which could be associated with unnoticed postoperative complications. 

Recently, in our hospital, this modification of the technique was adopted and was performed in the last 30 cases, obtaining a pain control rate of 97% and complication rate of less than 4%. In none of the cases, coagulation of Dandy’s vein was performed. The average surgical time was 60 min, from the incision up to the dural closure; average hospital stay was 1.5 days; and the mean postoperative recovery period was 5 days, with a direct impact on the patient’s quality of life and an immediate suspension of medications. Evaluating this, we consider the retrosigmoid parasterional burr-hole approach, along with the modification of the dural opening, as a versatile and effective technique for the management of vascular compression of the trigeminal nerve, with favorable results, a reduction in hospital stays and postoperative recovery, and minimal aesthetic alteration, placing this technique above the traditional retrosigmoid approach that entails a wide craniotomy with unnecessary exposure of the cerebellar cortex. 

## 5. Conclusions

Microvascular decompression (MVD) of the trigeminal nerve through a retrosigmoid parasterional burr-hole approach is a feasible and safe technique for managing the upper, middle, and lower cerebellar complexes without associated complications. Successful implementation of this technique requires a detailed understanding of the anatomy and landmarks involved. In spite of the small sample size, limited duration for comprehensive patient evaluation, and well-established nature of the technique employed, the significance of our study lies in its emphasis on the importance of employing clinometric scales for the assessment of pain and evaluation of treatment outcomes.

## Figures and Tables

**Figure 1 biomedicines-11-02707-f001:**
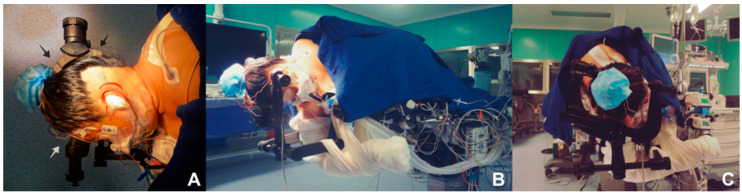
Position, fixation, and location. (**A**) Full lateral position of the head parallel to the ground; extension and cephalic flexion are avoided to favor venous return. The black (posterior) and white (anterior) arrows show the location of the park bench position reference points. (**B**) The contralateral arm is off the table to avoid brachial plexus injury. (**C**) The head is positioned with a three-point cephalic support to offer comfort for the approach.

**Figure 2 biomedicines-11-02707-f002:**
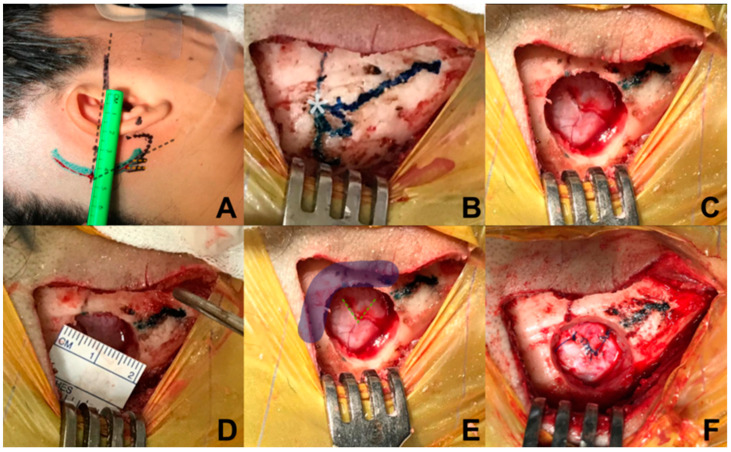
Marking and reference points. (**A**) Identification of the asterion projecting towards posterior the orbitomeatal line of Reid’s; (**B**) exposure of the digastric groove, base of mastoid, and asterion; (**C**) burr-hole with self-locking cutter on the lower and medial edge of the asterion; (**D**) the size of the burr-hole is 20 mm; (**E**) in order to allow visualization of the inferior border of the transverse sinus and medial border of the sigmoid sinus, (**F**) dural opening is performed in a triangular shape with a 7 mm base approximately towards the sigmoid sinus.

**Figure 3 biomedicines-11-02707-f003:**
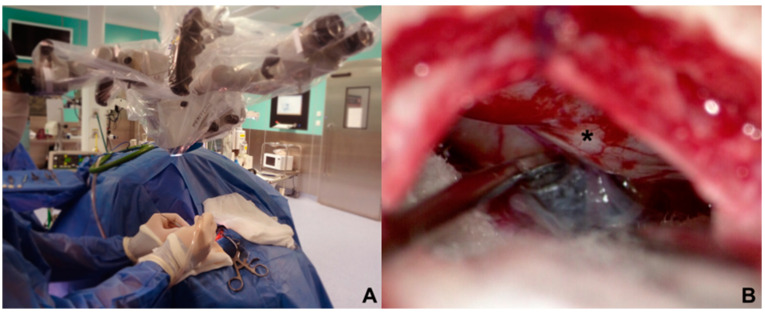
(**A**) Contralateral side rotation in parasterional burr-hole (**B**) improves visualization of the posterior petrosal face (*) that allows this surgical corridor to be developed without cerebellar retraction.

**Figure 4 biomedicines-11-02707-f004:**
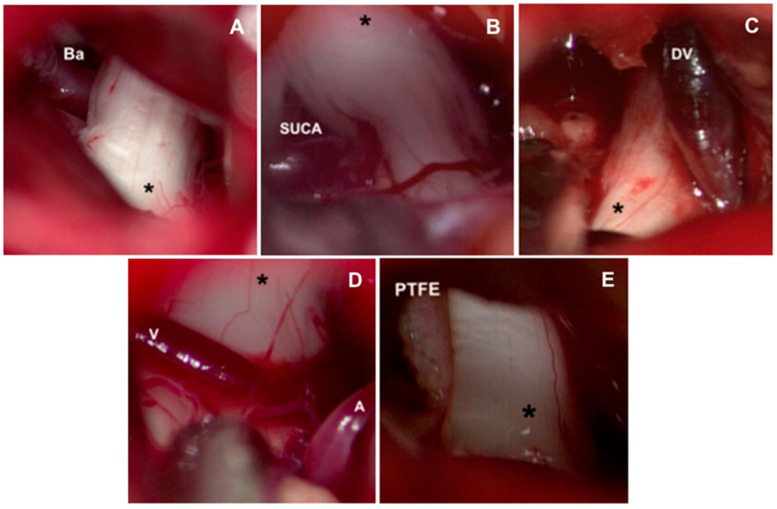
Causes of vascular compression in trigeminal neuralgia. (**A**) Basilar artery ectasia may be associated with trigeminal neuralgia. (**B**) The most common vessel identified for causing compression is the superior cerebellar artery. Veins are also identified for causing compression, finding frequently: (**C**) superior petrosal vein or (**D**) transverse pontine veins. (V: Vein, A: Artery). (**E**) Interposition of polytetrafluorethylene interface (PTFE) between the superior cerebellar artery and trigeminal nerve ganglion (*). Ba: Basilar artery. SUCA: Superior cerebellar artery. DV: Dandy vein.

**Figure 5 biomedicines-11-02707-f005:**
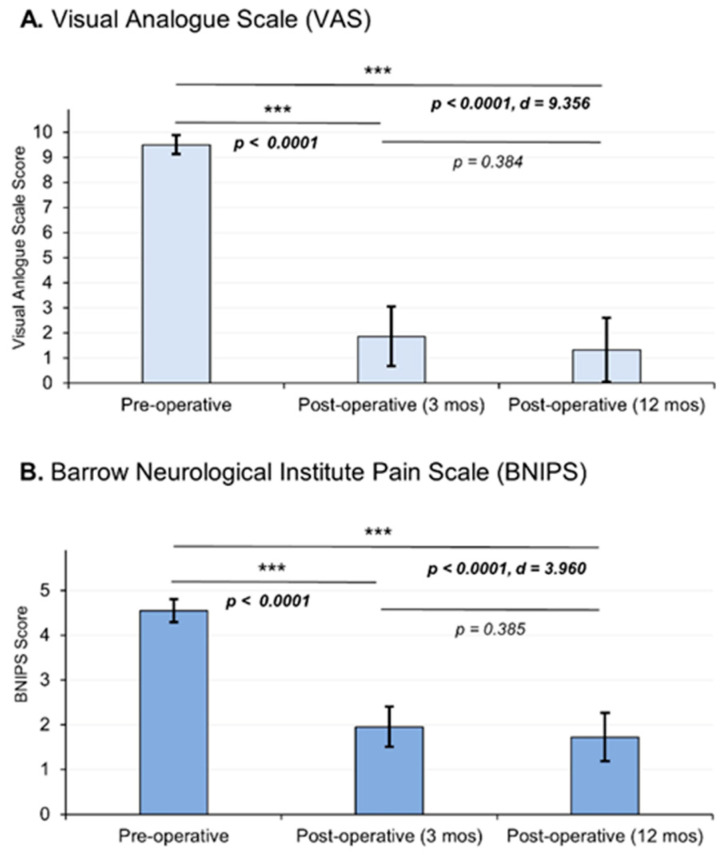
Postoperative pain outcomes were assessed, and the graphical representation illustrates the mean values and standard deviations. Statistical analysis was performed using an analysis of variance (ANOVA) test, followed by post hoc analysis employing either the DMS or C-Dunnett tests. The choice of post hoc test depended on the homogeneity of variances, as determined using Levene’s test. Effect size, measured using Cohen’s d, demonstrated a statistically significant reduction in pain intensity. The pain intensity was measured using two scales: the Visual Analogue Scale (VAS) (**A**) and the Barrow Neurological Institute Pain Scale (BNIPS) (**B**). Asterisks show the levels of significance.

**Table 1 biomedicines-11-02707-t001:** Clinical and demographical aspects of included patients.

*Sample size (n)*	22
*Demographic*	
Age (yrs)	54.9 ± 12.73
Males	4 (18.1%)
Females	18 (81.9%)
*Side affected*	
Right	19 (86.4%)
Left	3 (13.6%)
*Etiology*	
Arachnoiditis	4 (18.1%)
SCA	13 (59.1%)
VC	2 (9.1%)
BA	1 (4.6%)
EC	1 (4.6%)
AICA	1 (4.6%)
*Distribution*	
V1	3 (13.6%)
V2	0
V3	1 (4.6%)
V1/V2	3 (13.6%)
V2/V3	13 (59.1%)
V1/V2/V3	2 (9.1%)

SCA: Superior cerebellar artery. VC: Venous compression. BA: Basilar artery. EC: Epidermoid cyst. AICA: Anterior inferior cerebellar artery.

**Table 2 biomedicines-11-02707-t002:** Pain outcomes according to Visual Analogue Scale (VAS).

	Pain Evaluation		
VAS (Score)	Pre-Operative	Post-Operative (3 mos)	Post-Operative (12 mos)
10	14 (63.6%)	0	0
9	5 (22.7%)	1 (4.5%)	1 (4.5%)
8	3 (13.7%)	1 (4.5%)	1 (4.5%)
7	0	0	0
6	0	0	0
5	0	0	0
4	0	0	0
3	0	1 (4.5%)	2 (9.1%)
2	0	10 (45.4%)	3 (13.6%)
1	0	1 (4.5%)	0
0	0	8 (36.3%)	15 (68.3%)

VAS: Visual Analogue Scale. The outcomes are highlighted as a heatmap to show the main distribution of the patients at the different times of evaluation. The differences between shadows intensities are to show the frequency of the distribution of the data.

**Table 3 biomedicines-11-02707-t003:** Pain outcomes according to Barrow Neurological Institute Pain Scale (BNIPS).

	Pain Evaluation		
BNIPS (Score)	Pre-Operative	Post-Operative (3 mos)	Post-Operative (12 mos)
V	12 (55%)	0	0
IV	10 (45%)	2 (9%)	2 (9%)
III	0	2 (9%)	4 (18.4%)
II	0	11 (50%)	2 (9%)
I	0	7 (31.8%)	14 (63.6%)

BNIPS: Barrow Neurological Institute Pain Scale. The outcomes are highlighted as a heatmap to show the main distribution of the patients at the different times of evaluation. The differences between shadows intensities are to show the frequency of the distribution of the data.

## Data Availability

The data associated with the paper are not publicly available but are available from the corresponding author on reasonable request.

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
