# Peer review of "Minimally Invasive Retrosigmoidal Parasterional Burr-Hole Approach: Technique and Neuropathic Pain Amelioration after Microvascular Decompression of the Trigeminal Nerve"

_biomedicines, 2023, doi:10.3390/biomedicines11102707_

Round 1

Reviewer 1 Report

Thank you for the opportunity to review this manuscript.

The authors describe the surgical technique and clinical outcomes of microvascular decompression for trigeminal neuralgia using a minimally invasive retrosigmoidal parasterional burr-hole technique. The study demonstrates promising results in terms of pain relief and presents a feasible and safe surgical approach.  

The manuscript is relevant, well-written, and concise, and I have no significant concerns.

A suggestion to rewrite the conclusions would be: "Microvascular decompression (MVD) of the trigeminal nerve through a retrosigmoid parasterional burr-hole approach is a feasible and safe technique for managing the upper, middle, and lower cerebellar complexes without associated complications. Successful implementation of this technique requires a detailed understanding of the anatomy and landmarks involved. While the technique itself is well-known, the significance of our study lies in emphasizing the importance of using clinimetric scales for assessing pain and evaluating treatment outcomes."

Author Response

Dear Reviewer

We have incorporated changes based on the suggestions provided. We have highlighted the changes within the manuscript. Here is a point-by-point response to your  comments and concerns.

Comments:

Comment 1:” A suggestion to rewrite the conclusions would be: Microvascular decompression (MVD) of the trigeminal nerve through a retrosigmoid parasterional burr-hole approach is a feasible and safe technique for managing the upper, middle, and lower cerebellar complexes without associated complications. Successful implementation of this technique requires a detailed understanding of the anatomy and landmarks involved. While the technique itself is well-known, the significance of our study lies in emphasizing the importance of using clinimetric scales for assessing pain and evaluating treatment outcomes.

Response: We appreciate your comments, we decided to rephrase the conclusion considering your contributions (Line 266-272).

Sincerely

Dr. José Damián Carrillo-Ruiz.

Reviewer 2 Report

The paper addresses two entities - burr-hole retrosigmoidal approach to the pontocerebellar angle (1) and effectiveness of the microvascular decompression (2). However the effectiveness of decompression is well established and documented in the literature, as it is already stated in the discussion section by the authors. The surgical technique is the most interesting part of the work. The main idea is to emphasize, that adequate decompression could be performed through the minimally invasive opening. But there is no data on complications, duration and other possible reasons to choose this approach instead of standard one, presented. Consider revising the structure of the presentation, comparison with a standard retrosigmoid craniotomy must be included.

Figure 4 had no numeration of the pictures.

The english is poor, and must be corrected.

Author Response

Dear Reviewer

We have incorporated changes based on the suggestions provided. We have highlighted the changes within the manuscript. Here is a point-by-point response to your  comments and concerns.

Comments:

Comment 1: There is no data on complications, duration and other possible reasons to choose this approach instead of standard one, presented. Consider revising the structure of the presentation, comparison with a standard retrosigmoid craniotomy must be included.

Response: We appreciate your observation, we decided to add in the results section the complications observed after surgery (lines 191-196). On the other hand, due to the variety of names of the techniques, we decided to rephrase the first paragraph of the discussion, considering mentioning the relevant points that you mention to us (lines 198-211).

Comment 2: “Figure 4 had no numeration of the pictures. “

Response: Thanks for noting that mistake, we've added identifiers to the panels in Figure 4.

Comment 3: “The english is poor and must be corrected.”

Response: Your comment seems very appropriate to us, so we decided to send our manuscript to a language and style review with a native English speaker, some of the changes are highlighted in bold.

Sincerely

Dr. José Damián Carrillo-Ruiz.

Reviewer 3 Report

This paper is well written and explicative; description of the procedure and figures, as well discussion, are adequate. There are two main limitations of this study: the small number of cases and the short period of definitive evaluation of patients status assessment; the Authors have to highlight these limitations both in the abstract and in conclusion.

A minor editing of the English language is needed and a number of possible corrections to the text are advisable for better understanding. I have allocated a revised manuscript copy with side notes for completeness.

Author Response

Dear Reviewer

We have incorporated changes based on the suggestions provided. We have highlighted the changes within the manuscript. Here is a point-by-point response to your  comments and concerns.

Comments:

Comment 1: “There are two main limitations of this study: the small number of cases and the short period of definitive evaluation of patients status assessment; the Authors have to highlight these limitations both in the abstract and in conclusion.”

Response: We appreciate your observation, we decided to add the limitations that you mention to the conclusion both in the abstract (line 49), and at the end of the manuscript (266-272).

Comment 2:  “A minor editing of the English language is needed and a number of possible corrections to the text are advisable for better understanding. I have allocated a revised manuscript copy with side notes for completeness.”

Response: We greatly appreciate the time taken to make the corrections to the text, we modified and added all those points that were pointed out in the review manuscript copy with side notes.

Sincerely

Dr. José Damián Carrillo-Ruiz.

Round 2

Reviewer 2 Report

Adequate corrections were made, making study much more attractive. No more remarks. 

Author Response

Dear Reviewer

We have incorporated changes based on your suggestions. Thanks for your contribution. We look forward to hearing from you in due time regarding our submission and to respond to any further questions and comments you may have.

Sincerely

Dr. Armando Armas-Salazar and José Damián Carrillo-Ruiz.
